# Digital Economy Development, Common Prosperity, and Carbon Emissions: An Empirical Study in China

**Jingke Gao** [1] , **Wenxiao Zhou** [2], **Jinhua Cheng** [2,3] **and Ziyuan Liu** [4,*]

1 Institute of Advanced Study, China University of Geosciences (Wuhan), Wuhan 430074, China; gaojingke@cug.edu.cn

2 School of Economics and Management, China University of Geosciences (Wuhan), Wuhan 430074, China; wenxiaozhou@cug.edu.cn (W.Z.); chengjinhua100@126.com (J.C.)

3 Collaborative Innovation Center for Emissions Trading System Co-Constructed by the Province and Ministry, Wuhan 430205, China

4 School of Innovation and Entrepreneurship, Wuhan Institute of Technology, Wuhan 430205, China

* Correspondence: 23091105@wit.edu.cn

**Abstract:** Under the new development model, the digital economy has become a new engine to promote the green development of the economy and realize the goal of "double carbon". Based on the panel data of 30 provinces in China from 2010 to 2020, this paper empirically investigates the impact of the development of the digital economy on energy and carbon emissions using a series of econometric models such as baseline regression, a mechanism test, and the spatial Durbin model, etc. Common prosperity plays an intermediary role between digital economy development and carbon emissions; digital economic development optimizes resource allocation, effectively solves the problem of uneven resource distribution, and reduces energy and carbon emissions while achieving common prosperity. In addition, green innovation, industrial structure, urbanization level, R&D intensity, and the degree of marketization also have different degrees of influence on energy and carbon emissions. Therefore, the government should accelerate the construction of new digital infrastructure and implement the digital economy development strategy according to local conditions, so as to promote the digital economy to produce a more significant carbon emission reduction effect.

**Keywords:** digital economy; common prosperity; energy consumption; carbon emissions; spatial spillovers; heterogeneity

## 1. Introduction

In the last century, human economic activities immensely relied on non-renewable resources, which led to serious emissions of $CO_2$ as the main greenhouse gas. This has proven to have diverse effects on the life of humankind and the threat of the deteriorating sustenance of economic growth (Pata and Isik 2021). Report number five on the Intergovernmental Panel on Climate Change (IPCC) illustrates a discernible upward trend in the global average surface temperature. Over 90% of this increase is attributed to greenhouse gases produced by human activities, with $CO_2$ emissions from traditional energy consumption serving as a prominent contributor to the global greenhouse effect (Pata et al. 2023; Ahmed et al. 2022). In October 2021, China made a definitive commitment that, by 2030, $CO_2$ emissions should come to a peak, with the ultimate goal of achieving carbon neutrality by 2060. Subsequently, the Chinese and Americans issued the "China-United States Glasgow on Enhanced Climate Action in the 21st Century 2020 Joint Declaration". This declaration signifies their commitment to strengthening climate cooperation and collaborating to uphold the outcomes of the Paris Agreement. It underscores the common responsibilities of major countries in addressing climate change (Khan et al. 2022a; Usman et al. 2022).

Based on the endowment of energy resources in China, it is imperative to strategically and progressively advance carbon peak initiatives, foster a comprehensive energy revolu-

tion, diminish reliance on traditional energy sources, enhance the utilization of coal in a clean and efficient manner, and expedite new energy system planning and implementation. Active engagement in global climate change governance is essential in this endeavor (Kamal et al. 2021a; Yang et al. 2022). Consequently, the primary focus of future advancement is based on curtailing the utilization of conventional non-renewable sources while concurrently enhancing energy efficiency. The "Made in China 2025" initiative underscores the critical necessity to expedite technological advancements, structure the optimization of energy, and propel the environmentally sustainable advancement of society. This requires profound collaboration with all industries facilitated through the digitally based economy.

The further development of the digital economy continues to give birth to new Internet technologies, and the spread of digital technology applications has effectively reduced energy intensity and energy consumption per unit of output. In addition, digitalization has promoted the disintegration and reorganization of the modern industrial chain characterized by high efficiency, high output value, and low emissions, providing new ideas and feasible paths for reducing carbon emissions and promoting the green development of the modern industrial chain. The Chinese government is actively pursuing robust measures for $CO_2$ reduction, encompassing the reduction in the generation of power from plants that are coal sources (Mohsin et al. 2021; Tong et al. 2018), decreased energy consumption (Soytas et al. 2007; Valadkhani et al. 2019), accelerated integration of non-fossil energy sources (Dong et al. 2018; Godil et al. 2021), increased deployment of infrastructure that charges electric-based vehicles (Liu et al. 2021a), realization of energy transition goals (Li et al. 2022), and optimization of the structure of industries (Cheng et al. 2018). Concurrently, China grapples with the challenges of the advancement of the economy and prosperity. The importance of the use of digitalization is underscored by five-year plan number fourteen in optimizing and upgrading China's future systems of economy. In accordance with the Information and Communications Technology Academy of China (CAICT) in 2021, the digitally based economy of China attained an increase rate with a percentage of 9.7 in 2020, which is equivalent to 39.2 trillion, becoming the key in driving the stable growth of China's economy (CAICT 2020). Scholars have put into analysis the correlation between digitally based economic advancement and the emission of $CO_2$ using various approaches. For instance, some scholars claim that the digitally based economy can reduce the emission of $CO_2$ effectively, a phenomenon more pronounced in China (Li et al. 2021b; Ma et al. 2022) and some third-world countries (Chen 2021). Others have explored a correlation that is not based on the digitally based economy and $CO_2$, arguing that the effect of reducing emissions becomes efficient as the digitally based economy of China grows to levels that are higher (Wang et al. 2023; Li et al. 2021a; Li and Wang 2022b).

Consequently, it is imperative for the Government of China to meticulously take into consideration the effects of digitally based economic advancement on the reduction in the emissions of $CO_2$ in its future development strategies. Based on the above research, this paper uses data from 30 provinces in China from 2011 to 2020, adopts a multi-dimensional indicator system to quantify the level of development of the digital economy, applies a variety of spatial econometric models to test the impact of the digital economy on carbon emission reduction, explores the mechanism and effect of the digital economy on carbon emission reduction, analyzes the spatial effect of the digital economy on carbon emission reduction from the spatial and temporal perspectives, and analyzes the regional heterogeneity of this impact.

This paper's contributions are manifested in three areas: Firstly, it explores the construction of indicators for the digitally based economy. The methods and standards for measuring the level of economic advancement that is digitally based have not been unified. This paper incorporates policy and talent environments into the construction of indicators of the digitally based economy for a more comprehensive assessment. Secondly, the existing literature primarily explores the correlation between the digitally based economy and the emission of carbon from aspects such as the structure of industries, structure energy, and the intensity of energy. "Common prosperity" is socialism's core requirement and

the essential aspect of the modernization of China. Incorporating common prosperity into digitally based economic research and the reduction in carbon can fill research gaps. Thirdly, this paper employs baseline regression and the effects of intermediary and spatial Durbin models to study the impacts of the digitally based economy and the reduction in carbon in energy comprehensively. A heterogeneous analysis of the digitally based economy impacts on carbonic reduction in energy is also conducted.

This paper is organized into seven parts. The first part is the introduction, which provides the research background of the achievement of the goal of "dual carbon" and digitally based economy advancement. Based on this foundation, the research objectives and this paper's innovative aspects are outlined. A literature review is the second part which introduces the digitalized economy, the emissions of carbon, and the digitalized economy effects on the emissions of carbon. This report also explains the drawbacks of recent research. Thirdly, it consists of an analysis of theoretical mechanisms and research hypotheses, which makes assumptions about the correlation between the digitally based economy and emissions of carbon using three approaches. In section four, it outlines the design of the research, detailing the model, variables, and selected sources of information. In section five, it presents the empirical results, accompanied by endogeneity and robustness tests. Part six discusses mediation impacts, the impacts of spatial spillover, and analysis in a heterogenic way. Finally, the seventh section presents the conclusions and outcomes of the policy.

## 2. Literature Review

### 2.1. Research on Digital Economy

Since the advent of the digitally based economy, researchers at home and overseas have shown a growing interest in this area. Especially in the area of the urgency of the change of climate globally, academia has also been interested in the effects of the digitally based economy in terms of carbon. In current years, significant research has been conducted on the digitalized economy as a new form of economy. Studies have revealed that the digitally based economy has significant effects on the restructuring of industries and advancement of the economy in a high-quality way and is of very critical use in economic growth. Researchers have primarily looked at digitally based economic advancement from three perspectives. In terms of the indicators of digitalized economy measurement, the province-level digital economic index in China is based on the development of computerization, internetization, and the digitization of transactions (Chen and Wu 2022). Considering the new digitally based economic growth, we calculated a digitally based economic advancement index for 30 provinces of China from three aspects: infrastructure that is digitalized, digitally based industrial advancement, and digitalized leadership (Awan et al. 2022). The digital economy exerts its influence on firms at the microlevel as follows: First, it flattens the organizational structure of enterprises, makes production processes more flexible, and supports innovative and R&D model marketing (Chen et al. 2022). Secondly, the digitally based economy helps companies optimize their capital structure, and hence factor productivity increases (Pang et al. 2023), and efficiency and information acquisition are improved, which in turn contributes to the overall value of businesses (Llopis-Albert et al. 2021). At the macrolevel, researchers have demonstrated that leveraging extensive industrial data and intelligent analytics to establish networked manufacturing can genuinely transform raw data into meaningful information. This optimization of production processes enhances efficiency, thereby increasing enterprise value (Fernández-Portillo et al. 2020). Entrepreneurial activities serve as an endogenous driver of the growth of the economy, and the advancement of the digitally based economy can stimulate entrepreneurship by enriching entrepreneurs with resources, optimizing the allocation factor, and subsequently promoting the growth of the economy in a high-quality way. The digitally based economy in making energy advancements more efficient and unlocking the potential of energy savings is increasingly recognized. Against the

background of the growth of the economy and management of the environment, the importance of digital information is gradually emphasized (Ozturk and Ullah 2022).

*2.2. Research on Carbon Emissions*

In the emissions of carbon literature context, the focus is on the emissions of carbon dioxide measurement and their consequences. For instance, standards for detecting carbon emission levels across different industries are highlighted (Khan et al. 2022b). In accordance with the elements of the consumption of energy in China, eight sources of energy, coal and coke being included, are selected to evaluate the emissions of carbon dioxide over time in China (Kong et al. 2022). The emissions of carbon are primarily influenced by policies made by the government, urbanization, the structure of the population, and air pollution. The regional emissions of carbon dioxide can be reduced by the market through structural, technological, and configurational effects in accordance with the Carbon Dioxide Emissions Trading Pilot Policy (Ma and Zhu 2022). From the perspective of auditing by the government, the initial control of resources, which is performed naturally by high officials can support the energy industry, saving, and a reduction in the emissions of carbon dioxide by increasing the use of the capacities of industry (Wu et al. 2020). Empirical research and its context have been studied and examined from a perspective focused on the development of urbanization and data use of the panel cities which is at the prefecture level. The results indicate that the implications of population and urbanization on the productivity of carbon in cities range from strong to moderate. Once a certain threshold is crossed, the urbanization of a country no longer increases the carbon productivity of cities (Visconti-Caparrós and Campos-Blázquez 2022). From a demographic standpoint, based on provincial panel data, empirical findings suggest that the increase in the proportion of the working-age population primarily leads to carbon emissions through production and consumption (Zhang and Liu 2022). The study assesses the synergistic effects of reducing emissions of carbon and pollutants in the air in different environments in accordance with empirical data; a reduction in the pollution of the air synergistically impacts reducing both emissions of carbon and air contaminants (Shi et al. 2023).

In the field of exploration of the reduction in carbon emission pathways, experts have found that the digitally based economy not only plays a crucial role in the adjustment of the structure of industries in China and high-quality economic advancement but also occupies an important role in driving the transition between old and new drivers of the economy. Old and new economies also have the ability to significantly reduce the emission of carbon dioxide. The digitally based economy's influence on carbonic discharge in Chinese cities stems from the industrial Information and Communications Technology (ICT) expansion, which is in need of large inputs of intermediate products of carbon which are carbon made from ICT sectors (Zhou et al. 2019). Scientists are delving into the carbon-reducing significance of new digital generational technologies that are ubiquitous. Geographically, the digital economy has reduced emissions of carbon in some parts and had a remarkable impact on emissions of carbon in neighboring areas (Cheng et al. 2023). Based on empirical evidence, some researchers have come to conclude that the contribution of the digitally based economy to the performance of carbon is primarily local, with minimal impacts on surrounding regions (Zhang et al. 2022). The emerging digitally based economy, created by the integration of recent digitally based technologies with the financial sector, can help in the reduction in emissions by promoting the use of technologies in industries and the use of digitalization in industry (Chen 2022). In addition, some scientists have found that Internet advancements will reduce the pollution of the environment by improving the production and efficiency of energy. However, Li and Wang (2022a) argue that the digitally based economy's effects on carbonic discharge are not a simple linear correlation of either positive contribution or negative inhibition but rather a non-linear inverted U-shaped correlation where innovation efficiency in terms of innovation serves as a crucial mediator channel for the digitally based economy's impacts on carbonic discharge. It is also pointed out that the impact of the digital economy on carbon emissions shows a U-shaped non-linear

relationship that first rises and then falls, which corresponds to the fact that the digital economy empowers traditional industries and promotes transformation and upgrading (Zhu et al. 2022).

In short, digitalized economic advancement has remarkable effects on carbonic discharge reduction. However, there is still considerable scope for research on the digitalized economic effects on carbonic reduction. Currently, researchers are yet to agree on the digitalized economic indicators of measurement, and there is a lack of a comprehensive and systematic evaluation system, as indicated in Table 1. In this paper, the policy environment and talent environment are added to the construction of digital economy indicators for a more comprehensive measurement. Furthermore, achieving common wealth is an essential requirement of socialism, and the process of achieving common prosperity also involves the reduction in carbon emissions. On this basis, this study utilizes data from 30 Chinese provinces from 2010 to 2020. By constructing a system of evaluation that is comprehensive for the digitalized economy and public prosperity and employing various econometric spatial models, this study explores the techniques and the digitally based economy's effects on carbonic discharge reduction.

**Table 1.** Digital economy index system.

| Author(s) | Digital Economy Indicators |
|---|---|
| Chen and Wu (2022) | Computerization, Internet, Digitalization |
| Yi et al. (2022) | Digital industrialization, Industry digitization |
| Cheng et al. (2023) | Digital infrastructure, Industrial structure, Digital industrial scale, Technological innovation in the digital economy |
| Chen et al. (2022) | Mobile phone penetration rate, Internet penetration rate, Related industry output, Related industry employees |
| Li et al. (2021a) | Digital infrastructure, Internet development, Digital industry, Digital finance |

## 3. Mechanistic Theory of Analysis and Hypothetical Research

### 3.1. The Digitally Based Economic Impacts on Carbon Emissions

The digitalized economy leverages the collaboration and application of digitalized technologies in economic activities related to energy, environment, and other fields, playing a role in environmental leadership by promoting the conservation of energy and reduction in emissions. Relevant studies indicate that the digitally based economy, through the integration and innovation of digitalized technologies, coordinates the correlation between the supply of energy and demand, encourages technological innovation, and transforms consumption patterns, achieving both accelerated industrial digitization and reduced carbonic discharge (Han et al. 2020; Kamal et al. 2021a). The digitally based economy primarily contributes to carbonic discharge reduction as follows: First, it gives information technology support for the governance of the environment (Liu et al. 2021b). The use of huge amounts of information, cloud computing, and remote sensing technologies supports dynamic real-time monitoring by the government of air quality, emission pollution, and the carrying capacity of the environment. Secondly, the digital economy supports low-carbon technology firms, hence the enhancement of the efficiency of production factors and resource allocation (Alam et al. 2022; Li et al. 2021a). The digitally based economy can alter the products' nature, new value creation, and digitally based environmental shaping, enable businesses to gain new competitive advantages, and optimize resource allocation. Thirdly, the intrinsic advantages and basic features of the digitalized economy (Wang et al. 2021), such as the dissemination of cross-temporal information, the creation of data, and sharing, can reduce activities that are unnecessary, thereby reducing emissions of carbon. On this basis, the following hypothetic proposal was suggested:

**H1:** *The digitalized economy can lead to a reduction in energy-related carbonic emission intensification.*

*3.2. The Digitally Based Economy's Impacts on Carbonic Discharge through Its Effect on Common Prosperity*

Common prosperity is an intrinsic requirement of socialism and represents a key feature in China's context of modernization style. The "White Paper on the Facilitation of Common Prosperity through the Digital Ecological Industry" indicates that " the digitally based economic advancement is a useful path to achieve common wealth" (Li et al. 2021b). China has entered the stage of common prosperity coinciding with the era of the digitalized economy. The digitally based economy has a positive effect on common prosperity, characterized by its dynamic and non-linear nature. Achieving common prosperity requires enhanced resource allocation, reduced energy consumption, and consequent mitigation of energy-related carbon emissions (Chen and Zhang 2023). Common prosperity, possessing developmental, shareable, and sustainable attributes, necessitates the harmonious coexistence of social and natural environments, the same as the development that is balanced with production systems (Liu et al. 2023). Common prosperity can drive the advancement of a green economy, promote improvements in the environment ecologically, and encourage economic transition toward low-carbon models that can sustain the environment. Therefore, the process of achieving common prosperity inevitably influences carbon emissions (Zhao et al. 2023). High-quality economic advancement, the narrowing of income disparities, and the development of a common economy contribute to the government's better formulation of low-carbon environmental policies (Zhang and Qian 2023). On this basis, the following speculation is proposed:

**H2:** *The digital economy affects energy and carbon intensity by optimizing the allocation of resources, improving the structure of social wealth distribution, and contributing to the development of common prosperity.*

*3.3. The Spatial Spillover Effects of the Digitally Based Economic Impacts on Carbonic Emissions*

Proximity enhances the elemental flow efficiency, establishing a closer correlation in nearby regions in relation to production, operation, and economic advancement (Ren et al. 2021). An essential feature of the digitalized economy is the spreading of technology worldwide and in different geographical spaces over time. Numerous factors affect the spatial effects of the digitally based economy, including the geographical interval, information acquisition, learning capability communication and transportation, and information acquisition, impacting the spatiotemporal trajectory of digital technology adoption and dissemination (Sobaci and Eryigit 2015). The spatial diffusion speed of digital technology varies, leading to distinct adoption curves in different regions. Additionally, the technological advancement process may exhibit a siphon effect. Regions prioritizing the development of digital industry sectors might transfer low-end, high-energy-consuming industries to economically underdeveloped areas, causing lagging regions to postpone the adoption of digital technology. However, during the technology diffusion process, a demonstration effect can also emerge. When a city effectively promotes its level of advancement, carbon emissions are reduced, and it becomes a leading region through vigorous digital economic development; its development model, policy framework, and innovative experiences influence neighboring and related provinces, generating demonstration and learning effects. This contributes to the adjustment of local policies or economic activities in surrounding regions (Yi et al. 2022). On this basis, the following hypothetical research proposal is suggested by this study:

**H3:** *The digitally based economy's advancement has many spatial effects on carbonic emissions.*

Existing studies have provided a foundational assessment and impact analysis framework mechanisms through which the digitally based economy promotes the reduction in carbonic discharge. This study categorizes the influence of the digitally based economy on the energy-related reduction in carbon into three major types: direct effects, indirect effects,

and excess spatial effects. The techniques through which the digitalized economy affects the energy-related reduction in carbon are illustrated in Figure 1.

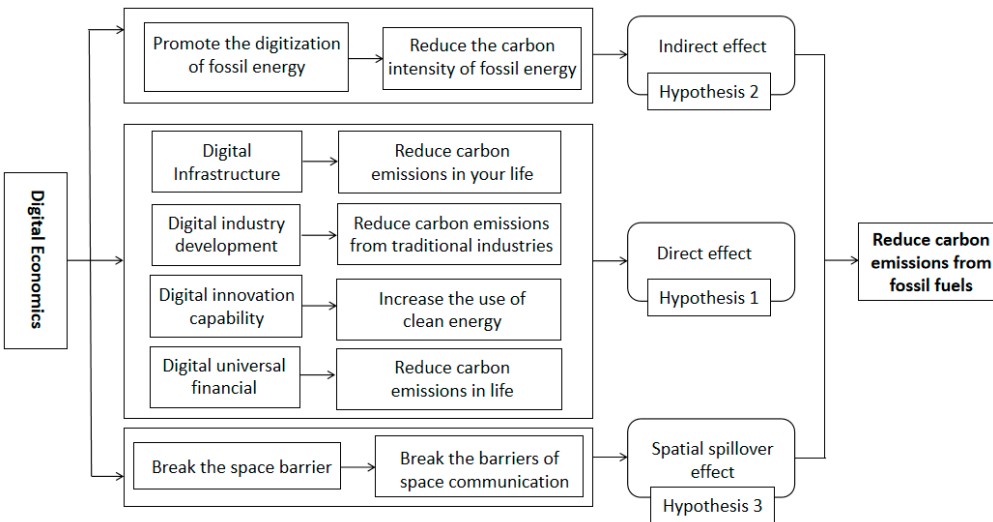

**Figure 1.** Conceptual framework.

## 4. Model and Data

*4.1. Variable Description*

### 4.1.1. Explained Variables

Energy Intensity of Carbonic Emission (CEI). CEI is indicated by the carbonic emissions (CEQ) per unit of

$$CEI = \frac{CEQ}{GDP} \qquad (1)$$

This paper delves into energy-related carbon emission reduction, and energy includes eight major sources: diesel, fuel oil, liquefied petroleum gas coke, crude oil, gasoline, kerosene, and natural gas. Other energy sources are not considered due to their small proportion and data deficiencies. The method employed for calculating the emissions of carbon from the consumption of energy follows the IPCC(The Intergovernmental Panel on Climate Change)uidelines (Guo et al. 2022), and its calculation formula is as follows:

$$CEQ = \sum_{i=1}^{8} CO_{2,i} = \sum_{i=1}^{9} E_i \times NCV_i \times CEF_i \times COF_i \times \frac{44}{12} \qquad (2)$$

In the equation, CEQ represents carbon emissions; $E_i$ and $NCV_i$ respectively denote the consumption and lower heating value of various types of energy. $CEF_i$ stands for the oxygen content of various energy sources; $COF_i$ is the carbon emission factor for various fossil fuels; 44 and 12 represent the chemical molecular weights of carbon dioxide and carbon, respectively.

### 4.1.2. Explanatory Variables

Digitally based economy (DIG). In measuring the level of digitally based economy advancement, based on mechanism analysis and incorporating previous research findings (Nicolay 1999), this study performs an assessment based on five dimensions: digitalized infrastructure, digitally based industrial advancement, digitally based innovation capability, digitally based financial inclusivity, and the digital advancement of environment. This forms platforms for evaluating the level of digitally based economy development, as indicated in Table 2. In terms of measuring the digitally based economy advancement level, the entropy weight method is employed for calculation (Zhu et al. 2022).

**Table 2.** Measurement index system of the digital economy.

| Primary Indicators | Secondary Indicators | Description of Indicators (Units) | Index Attribute |
|---|---|---|---|
| Infrastructure | Broadband Internet infrastructure | Internet users (10,000 households) | + |
| | Mobile Internet fundamentals | Mobile telephone subscribers (10,000 households) | + |
| Industry development | E-commerce Industry development | Urban e-commerce parks (10,000) | + |
| | Infrastructure of the information industry | Employees in information transmission, Computer services and software (10,000 persons) | + |
| | Telecommunications industry output | Total telecommunication business (100 million) | + |
| Innovation capacity | Digital innovation factor support | Expenditure on science and technology (100 million) | + |
| | Level of digital Innovation output | Patents related to the digital economy (10,000) | + |
| | Digital Hi-Tech penetration | Degree of penetration of digital high-tech applications among listed companies (%) | + |
| Inclusive finance | Digital inclusive finance | Digital financial inclusion coverage breadth index (%) | + |
| | Depth of use | Depth of use index for digital financial inclusion (%) | + |
| | Degree of digitization | Digital financial inclusion digitization index (%) | + |
| Development environment | Talent environment | Number of graduates from general higher education institutions (10,000 persons) | + |
| | Policy environment | Internal expenditure on R&D personnel costs (100 million) | + |

### 4.1.3. Intermediary Variable

Level of common prosperity (CP). The common use of "common" and "prosperity" forms the basis of common wealth. Therefore, this paper gives a measurement of the level of common prosperity in Chinese provinces from two aspects: "common" and "prosperity" (Liu et al. 2023), as indicated in Table 3. The technique of entropy is used to determine the common prosperity level.

**Table 3.** Common prosperity development level indicator system.

| First Level Index | Second Level Index | Third Level Index | Impact |
|---|---|---|---|
| Prosperity | Resident life | GDP per capita | + |
| | | Per capita disposable income of rural residents | + |
| | | Per capita consumption expenditure of urban residents | + |
| | | Per capita consumption expenditure of rural residents | + |
| | Education level | Per pupil education expenditure | + |
| | Medical level | Number of hospital beds per capita | + |
| | Social service level | Local fiscal general budget expenditure/GDP | + |
| | Cultural life level | Public library holdings per capita | + |
| | Science and education input | Science and education expenditure/GDP | + |

**Table 3.** *Cont.*

| First Level Index | Second Level Index | Third Level Index | Impact |
|---|---|---|---|
| Commonality | Urban–rural gap | The ratio of rural residents' income to urban residents' income | + |
| | | The ratio of consumption expenditure of rural residents to that of urban residents | + |
| | Regional gap | The ratio of rural residents' income to the national average rural residents' income | + |
| | | The ratio of urban residents' income to the national average urban residents' income | + |

#### 4.1.4. Control Variables

To evade biases in empirical test results resulting from overlooking crucial explanatory variables, the paper introduces the following five control variables into the model: (1) Green Technology Innovation (INNOV) (Zhang et al. 2023); (2) Industrial Structure (IS) (Xue et al. 2022); (3) Degree of Marketization (MAR) (Tan and Kaili 2023); (4) Research and Development Intensity (INUR) (Liu et al. 2017); and (5) Urbanization Level (UR) (Chen and Lin 2021). Table 4 provides the definitions and descriptions of the variables involved while Table 5 shows the statistics of the variables in a descriptive manner.

**Table 4.** Control variable selection and measure.

| Variable | Symbol | Name | Measure |
|---|---|---|---|
| Dependent variable | CEI | Energy carbon intensity | Carbon emissions per unit of GDP |
| Explanatory variable | DIG | Development level of the digital economy | Calculated according to the index system of the digital economy |
| Intermediary variable | CP | Development level of the mutual enrichment | Calculated according to the index system of the common wealth |
| Control variables | INNOV | Green technology innovation | Number of green patents granted |
| | IS | Industrial structure | Ratio of tertiary to secondary sector |
| | MAR | Marketability | Regional marketization index |
| | INUR | R&D intensity | Ratio of R&D expenditure to GDP |
| | UR | Urbanization level | Ratio of urban population to total population |

**Table 5.** Descriptive statistics of variables.

| Variable | Obs | Min | Mean | Max | Std. Dev. |
|---|---|---|---|---|---|
| CEI | 330 | 0.325 | 2.306 | 9.098 | 1.748 |
| DIG | 330 | 0.049 | 0.238 | 1.000 | 0.182 |
| CP | 330 | 0.187 | 0.763 | 2.245 | 0.436 |
| INNOV | 330 | 3.583 | 7.331 | 10.450 | 1.351 |
| IS | 330 | 0.518 | 1.218 | 5.296 | 0.695 |
| MAR | 330 | 0.526 | 6.804 | 11.910 | 2.056 |
| INUR | 330 | 0.004 | 0.016 | 0.011 | 0.064 |
| UR | 330 | 0.350 | 0.590 | 0.896 | 0.122 |

#### 4.2. Model Construction

Base of Regression Test. Firstly, to examine research hypothesis H1 proposed in the analysis in terms of the theory above, the paper constructs the following baseline regression model based on the direct mechanism transmission of the digitally based economy to intensity carbonic discharge:

$$\text{CEI}_{\text{it}} = \alpha_0 + \alpha_1 \text{DIG}_{\text{it}} + \alpha_n X_{\text{it}} + \mu_i + \delta_i + \varepsilon_{\text{it}} \tag{3}$$

Baseline Regression Test. In the equation, i stands for the sample province; t denotes the annum; $CEI_{it}$ indicates carbonic discharge intensification; $DIG_{it}$ stands for the digitally based economy; $X_{it}$ represents control variable series; $\mu_i$ is the individually fixed impact of the province, which does not vary over time; $\delta_i$ fixed effects of control time; and $\varepsilon_{it}$ is the random disturbance term.

Mediation Mechanism Test. Additionally, the direct effects presented by the baseline regression model (1), to explore the potential indirect mediation techniques of the digitally based economy on the intensity of carbonic discharge, on the basis of research hypothesis H2 stated above, this study refers to the application of the mediation effects model by (Adner et al. 2019). The specific testing method is as follows: based on the significant estimation coefficients of the digitalized economy in the baseline regression model (1) for the intensity of carbon emission, we establish a regression equation for the digitally based economy on the intermediate variable (M) and a linear regression equation for the digitalized economy and the intermediate variable (M) on the intensity of carbon emission. For this purpose, the form which is specific to the mediation effect model of regression is set as follows:

$$M_{it} = \beta_0 + \beta_1 DIG_{it} + \beta_n X_{it} + \mu_i + \delta_i + \varepsilon_{it} \tag{4}$$

$$CEI_{it} = \gamma_0 + \gamma_1 DIG_{it} + \gamma_2 M_{it} + \gamma_n X_{it} + \mu_i + \delta_i + \varepsilon_{it} \tag{5}$$

Spatial Durbin Model (SDM) Test. Finally, to further explore whether the effects of a digitally based economic reduction in carbonic discharge exhibit a spatial spillover phenomenon, spatial interaction terms corresponding to a digitally based economy, the intensification of carbonic discharge, and a series of variables of control are added to the baseline regression model (1). This extends the model into an SDM (Abbas et al. 2021), with the specific form as follows:

$$CEI_{it} = \alpha_0 + \alpha_1 DIG_{it} + \alpha_n X_{it} + w_{it}(\rho_1 CEI_{it} + \rho_2 DIG_{it} + \rho_n X_{it}) + \mu_i + \delta_i + \varepsilon_{it} \tag{6}$$

$\rho_1$ represents the spatial autoregressive coefficient, and $\rho_2$ and $\rho_n$ represent the coefficients of the spatial interaction terms for DIG and control variables; W denotes the spatial weight matrix. To ensure the effectiveness of empirical test results, the paper employs an economic geography distance matrix for regression (Elhorst 2010).

### 4.3. Data Sources

The article conducts an empirical study utilizing sample data from the mainland of Chinese 30 provinces, Tibet, Hong Kong, Macao, and Taiwan being excluded, spanning the years 2010 to 2020. The information utilized in this research includes digital financial data sourced from the Digital Financial Inclusivity Index published by Peking University's Digital Finance Center. Other data sources include China Environmental Yearbook, China Energy Statistics Yearbook, and National Bureau of Statistics. To mitigate potential interference from the non-stationarity of macrolevel data on empirical results, all variables employed in this paper undergo logarithmic transformations.

## 5. Empirical Results and Analyses

### 5.1. Analysis of Baseline Regression Results

According to the econometric model constructed in the previous section, the paper studies the digitally based economy's impacts on the intensification of carbonic discharge, as shown in Table 6. In the baseline regression tests, regression estimates are conducted by separately including and not including the regression model which has control variables. Simultaneously, provincial and yearly fixed effects are employed to enhance the robustness of the regression estimates (Wang et al. 2022b). The coefficient that was estimated on digitally based economy advancement on the intensification of carbonic discharge is negatively remarkable in the baseline regression, and the inclusion of control variables does not change the direction of the estimated coefficient. This indicates that the advancement of the digitalized economy has an inhibitory effect on the intensification of carbonic discharge,

providing ample evidence for the digitalized economic effects on carbonic reduction, thus confirming Hypothesis 1.

**Table 6.** Analysis of baseline regression results.

| | (1) CEI | (2) CEI | (3) 2SLS |
|---|---|---|---|
| DIG | −2.493 *** (−4.81) | −3.953 *** (−5.26) | −1.655 *** (−2.63) |
| INNOV | | −0.317 *** (2.72) | −0.449 ** (0.072) |
| MAR | | −0.136 ** (−2.93) | 0.039 * (−0.111) |
| IS | | 0.108 (0.23) | 0.395 ** (0.365) |
| INUR | | 29.835 ** (2.05) | 1.186 ** (0.094) |
| UR | | −9.625 *** (−6.51) | −0.395 (0.175) |
| _Cons | 3.246 *** (15.89) | 8.023 *** (2.72) | −14.797 ** (1.632) |
| Observations | 330 | 330 | 300 |
| Province FE | YES | YES | YES |
| Year FE | YES | YES | YES |
| Kleibergen-Pa ap rk LM statistics | | | 89.833 [0.00] |
| Kleibergen-Pa ap Wald rk F statistics | | | 56.473 [0.00] |
| $R^2$ | 0.066 | 0.964 | 0.607 |

Note: ***, **, and * indicate significance at the 1%, 5%, and 10% levels, respectively. T-statistics are in parentheses.

The indicators for INNOV all pass negative significance tests, suggesting that an increase in the number of applications for green patents is conducive to promoting the gradual replacement of coal and oil with clean energy, thereby reducing carbon emissions. The indicators for MAR all pass negative significance tests, indicating that an increase in marketization level improves resource allocation efficiency, reduces pollution emissions per unit output, and promotes low-carbon development. The indicators for IS all pass positive significance tests, indicating that the current industrial structure is unconducive to the discharge of carbon, possibly due to an unreasonable structure of industries. The indicators for INUR all pass positive significance tests, possibly because they focus on breakthroughs in zero-carbon industrial processes, carbon capture and utilization, and storage, reducing carbon emissions. The indicators for UR all pass negative significance tests, possibly because the urbanization level has its positive externalities, promoting carbon reduction processes through scale effects, resource allocation effects, and other pathways.

*5.2. Endogeneity Test*

Considering the potential endogeneity of the empirical results of the digitalized economy on the emissions of carbon which could impact the reliability of the conclusions in this study, a two-stage least squares method is employed for endogeneity testing. In addition, this study attempts to mitigate the potential reverse causality issue between the digitally based economy and energy-related emissions of carbon using the instrumental variable approach. Following the approach (Bai et al. 2023; Huang et al. 2023), the chosen instrumental variable is the distance that is spherical to Hangzhou from provincial capitals. The reasons are that the birthplace of digital financial platforms like Hangzhou and Alipay is leading the development of the digitally based financial sector. It can be expected that the smaller the geographical distance to Hangzhou, the higher the digitally based

economy's advancement. Additionally, geographical interval, as a natural feature that is geographically typical, is not closely correlated to carbonic discharge, meeting the exogenic requirement. For empirical analysis, this study constructs terms of interaction between variables that are instrumental and the ones that are yearly dummies. The results are indicated in Table 5, column (3), with the coefficient of the regression being significant.

*5.3. Robustness Test*

5.3.1. Replace the Variables That Are Core Explanatory

Drawing on Lin and Li's (2011) study, this study selected energy-related discharge of carbon as the robustness dependent variable testing. The outcomes are shown in Table 6, column (3), with a significant regression coefficient. This indicates that, even when replacing the dependent variable, the digitally based economic advancement continues to reduce the energy-related discharge of carbon significantly in various provinces.

5.3.2. Replace the Explained Variable

In the earlier stages, empirical regression on carbon emissions was conducted using the Digitally based Economic Index based on the technique of entropy weight. In the robustness test, this index was replaced with the Digitally based Economic Index obtained through the component of principal analysis (Wang and Zhong 2023). The outcomes, tabulated in Table 6, column (4), reveal a highly significant regression coefficient. The digitally based economy reduces the discharge of carbon significantly, aligning with the regression results in the preceding sections and demonstrating robustness in the findings.

5.3.3. Replacement Sample Period

This study employs a robustness test by shortening the sample period, following the approach of Xing et al. (2023). After 2015, China experienced rapid development in the digital economy, and literature uses 2015 as the starting year for research in the area of a digitally based economy. In this study, data were obtained from 2015 to 2020 for empirical estimation. The results, shown in Table 7, column (5), still exhibit a significantly negative regression coefficient, consistent with the baseline regression results of this study, validating the hypotheses presented in this paper.

**Table 7.** Robustness tests.

| | (1) | (2) | (3) | (4) | (5) |
|---|---|---|---|---|---|
| DIG | −2.493 *** (−4.81) | −3.953 *** (−5.26) | −0.724 ** (−2.88) | −0.811 *** (−5.68) | −7.768 * (−2.86) |
| INNOV | | −0.317 *** (2.72) | −0.139 *** (3.71) | −0.266 * (−2.05) | −0.253 (−0.45) |
| MAR | | −0.136 ** (−2.93) | −0.0398 * (−2.53) | −0.125 ** (−2.70) | −0.183 (−1.93) |
| IS | | 0.108 (0.23) | −0.123 * (−2.26) | 0.251 (1.50) | −1.748 ** (−3.25) |
| INUR | | 29.835 ** (2.05) | 6.874 (−1.39) | 37.79 * (2.54) | 89.100 (1.59) |
| UR | | −9.625 *** (−6.51) | −0.654 (1.10) | −9.495 *** (−5.73) | −21.71 (−1.64) |
| _Cons | 3.246 *** (15.89) | 8.023 *** (2.72) | 9.505 *** (12.64) | 7.803 *** (3.48) | 25.63 (1.67) |
| Province FE | YES | YES | YES | YES | YES |
| Year FE | YES | YES | YES | YES | YES |
| Observations | 330 | 330 | 330 | 330 | 330 |
| $R^2$ | 0.006 | 0.964 | 0.977 | 0.965 | 0.993 |

Note: ***, **, and * indicate significance at the 1%, 5%, and 10% levels, respectively. T-statistics are in parentheses.

## 6. Discussion

### 6.1. Analysis of Intermediary Mechanisms

In the analysis in terms of theory in the previous section, the role of the transmission mechanism of the digitally based economy bringing effect on the intensification of carbon discharge of various provinces was analyzed from the perspective of the intensification of energy consumption. To verify this assumption of the transmission mechanism, the paper borrows the method from Tsurumi and Managi (2010) and employs a mediation effect test for empirical analysis, with the outcome of the test tabulated in Table 8. In the first column, the coefficient for DIG is significantly negative, meeting the primary condition of the mediation effect test. The results in the second column show that the digitally based economic advancement reduces the intensification of energy consumption significantly. Observing the coefficient, with control variables held constant, the digitally based economic increase by 1% can lead to a 1.817% decrease in energy consumption intensity. The results in the third column indicate that the intensity of energy consumption is correlated to the discharge of carbon intensity positively. For every 1% decrease in energy consumption intensity, the intensity of carbon discharge can be reduced by 2.553%, demonstrating the existence and significance of the mediation effect.

**Table 8.** Regression models for intermediate variables.

| | (1) CEI | (2) CEI | (3) CEI |
|---|---|---|---|
| DIG | −2.493 *** (−4.81) | −1.817 *** (−5.29) | −1.721 (−1.49) |
| CP | | | 2.553 *** (13.45) |
| INNOV | | −0.118 *** (4.12) | −0.017 (0.18) |
| MAR | | −0.0330 *** (−2.89) | −0.052 (−1.41) |
| IS | | 0.005 * (1.66) | −0.01 (−1.03) |
| INUR | | 15.716 *** (4.42) | −10.284 (−0.88) |
| UR | | −3.033 *** (−8.38) | −1.883 (−1.46) |
| _Cons | 3.246 *** (15.89) | 1.743 *** (3.20) | 3.573 ** (2.01) |
| Province FE | YES | YES | YES |
| Year FE | YES | YES | YES |
| Observations | 330 | 330 | 330 |
| $R^2$ | 0.066 | 0.967 | 0.978 |

Note: ***, **, and * indicate significance at the 1%, 5%, and 10% levels, respectively. T-statistics are in parentheses.

For the robustness guarantee of the mediation effect, this research conducted a bootstrap test (Wu et al. 2021), as shown in Table 9. It can be observed that the effects that are indirectly and directly correlated to energy consumption intensity are important. The 95% confidence interval for the indirect effect is [3.423–4.264], excluding 0, indicating the presence of a mediation effect.

**Table 9.** Bootstrap test of energy structure.

|  | Estimated Coefficient | Standard Error | Z Value | 95% Confidence Interval | |
|---|---|---|---|---|---|
| Direct Effect | 0.387 *** | 0.251 | 0.231 | 0.128 | 0.246 |
| Indirect Effect | 3.843 *** | 0.214 | 7.811 | 3.423 | 4.264 |

Note: ***, indicates significance at the 1%, 5%, and 10% levels, respectively. T-statistics are in parentheses.

### 6.2. Spatial Spillover Analysis of Effects

Before an analysis of economically centered spatial effects is conducted, it is necessary to separately perform an investigation on the existence of spatial effects for both the digitally based economy and the carbon discharge intensity. In this regard, Moran's I index method (Wen et al. 2020) is used to test the spatial autocorrelation on both factors, and the outcomes are tabulated in Table 10. The outcomes of the test indicate that the indices of Moran I for the digitally based economic advancement index and intensity of carbonic discharge are significant from 2010 to 2020. This indicates a clear spatial autocorrelation for the digitally based economy and carbon discharge intensity in all provinces during the study period, reflecting the spatial clustering distribution characteristics of both. The use of an economically centered spatial analysis is therefore considered reasonable and feasible. Furthermore, it is necessary to select an appropriate spatial econometric model for empirical analysis. Following the methodology described in Bai et al. (2023) to determine spatial econometric models, LM tests, Hausman tests, LR estimates, and Wald tests are performed sequentially. All outcomes of the tests indicate that the spatial model of Durbin with dual fixed impacts is optimal. Table 10 presents the outcomes of the spatial model of Durbin (SDM), which tests the effects of the digital economy on the intensity of carbon discharge within the economic geographic interval matrix of the spatial weight matrix. Examining the results, the spatial autoregressive coefficient ($\rho$) for discharge of carbon intensity is significantly positive at the 1% level under both matrices of spatial weights. This means that there are regional spillover effects on the discharge of carbon dioxide intensity, where the increase in the intensity of emission of neighboring counties worsens the emission intensity of the focus area. To analyze the digitally based economy's advancement impacts that are both direct and indirect on the reduction in the discharge of carbon dioxide, the partial disaggregation technique suggested by Wang et al. (2022a) is utilized in the decomposition of the overall regional effect of the digitally based economy on the intensity of carbon. The impact decomposition results presented in Table 11 below and the spatial weight matrix of economic geography show that the estimated coefficients for the indirect impact, direct impact, and total impact of the digital economy are all remarkably negative at the level of 1%. This suggests that the digitally based economy can effectively curb the intensity of the carbon dioxide discharge of the focus area and neighboring provinces, which will accelerate the growth of digitally based economic advancement and help reduce the emission intensity of carbon dioxide.

**Table 10.** Moran's I values of carbon emission intensity.

|  | 2010 | 2011 | 2012 | 2013 | 2014 | 2015 | 2016 | 2017 | 2018 | 2019 | 2020 |
|---|---|---|---|---|---|---|---|---|---|---|---|
| CEI | 0.354 *** | 0.352 *** | 0.354 *** | 0.342 *** | 0.340 *** | 0.303 ** | 0.316 ** | 0.353 *** | 0.354 *** | 0.389 *** | 0.316 ** |
|  | (3.266) | (3.316) | (3.331) | (3.241) | (3.221) | (2.919) | (3.003) | (3.328) | (3.364) | (3.674) | (3.007) |
| DIG | 0.110 * | 0.220 ** | 0.172 ** | 0.10 * | 0.140 * | 0.143 * | 0.917 ** | 0.197 ** | 0.204 ** | 0.261 *** | 0.269 *** |
|  | (1.381) | (2.339) | (1.983) | (1.722) | (1.680) | (1.715) | (2.176) | (2.126) | (2.219) | (2.727) | (2.822) |

Note: ***, **, and * indicate significance at the 1%, 5%, and 10% levels, respectively. T-statistics are in parentheses.

**Table 11.** Spatial Durbin model regression results.

| | (1)<br>CEI | (2)<br>Direct Effect | (3)<br>Indirect Effect | (4)<br>Total Effect |
|---|---|---|---|---|
| DIG | −3.032 ***<br>(−4.31) | −2.899 ***<br>(−3.60) | −1.170<br>(−0.50) | −4.069 **<br>(−1.98) |
| _cons | Yes | Yes | Yes | Yes |
| Province FE | Yes | Yes | Yes | Yes |
| Year FE | Yes | Yes | Yes | Yes |
| $\rho_1$ | −1.374 ***<br>(−5.56) | / | / | / |
| Sigma2_e | 0.084 ***<br>(12.27) | / | / | / |
| $R^2$ | 0.086 | 0.072 | 0.678 | 0.279 |

Note: ***, ** indicate significance at the 1%, 5% levels, respectively. T-statistics are in parentheses.

### 6.3. Heterogeneity Analysis

### 6.3.1. Heterogeneity in Different Regions

From Table 12, it is possible to observe that there is a disparity in terms of region in the digitally based economy's impacts on the emission of carbon dioxide in different parts. The reduction in carbon effects of the digitalized economy in the regions in the eastern and central parts is in line with the national level, while the emission intensity of carbon dioxide in the western areas is not remarkable. This may be due to the relatively backward development of digital infrastructure in the western region, leading to poor integration of the digital economy with green technologies, resulting in insignificant impacts on carbon reduction (Zeng and Yang (2023)). Therefore, in building China's digitalized economy, more attention must be paid to integrating regional digitalization with the aim of narrowing the gap in digitalized economic advancement between regions.

**Table 12.** Heterogeneity analysis.

| | (1) | | | (2) | | |
|---|---|---|---|---|---|---|
| | East | Central | Western | Developed | Medium<br>Developed | Less<br>Developed |
| DIG | −1.710 ***<br>(−3.38) | −6.994 *<br>(−4.68) | −4.951<br>(−2.14) | −0.913 **<br>(−3.62) | −1.490 *<br>(−2.27) | −0.845<br>(−2.25) |
| Control<br>variable | Yes | Yes | Yes | Yes | Yes | Yes |
| _Cons | 6.759 ***<br>(3.73) | 3.599<br>(1.41) | 6.69 **<br>(3.02) | 7.265 ***<br>(4.38) | 0.246<br>(0.06) | 4.140<br>(2.18) |
| Province FE | YES | YES | YES | YES | YES | YES |
| Year FE | YES | YES | YES | YES | YES | YES |
| $R^2$ | 0.972 | 0.982 | 0.962 | 0.993 | 0.944 | 0.981 |

Note: ***, **, and * indicate significance at the 1%, 5%, and 10% levels, respectively. T-statistics are in parentheses.

### 6.3.2. Heterogeneity in Levels of Economic Development

To examine the effects of the digitally based economy on the emission of carbon in provinces with different features of economic development, we draw on the research of scholars (Song et al. 2020). Using provincial economic advancement level as a dividing criterion and per capita GDP as a measure of economic advancement, we conducted group regression by calculating the median for the samples. The results are indicated in Table 11. The digitally based economy's effects on developed and moderately developed regions are more significant. This may be due to the scale advantages of developed regions in terms of the structure of industries, governance of environmental investment, and technological innovation, leading to certain agglomeration effects that facilitate resource optimization. In contrast, the effects are not significant in underdeveloped regions.

## 7. Conclusions and Policy Implications

### 7.1. Conclusions

On the basis of panel data from Chinese regions between 2010 and 2020, this paper focuses on typical facts indicating that the advancement of the digitally based economy can seriously affect changes in the intensity of carbon emission. Simultaneously considering the inherent attributes of the digitalized economy, the paper analytically examines the direct impacts and transmission techniques of the digitally based economy on the intensity of carbonic discharge from a theoretical perspective. By employing double fixed effects, mediating effects, and spatial Durbin models, the paper validates, from a two-dimensional to a three-dimensional perspective, the impact and mechanisms of the digitally based economy on the intensity of carbon emission. This research paper mainly concludes as follows: First of all, during the study period, China's digitalized economy achieved relatively rapid advancement, with obvious advantages in the digitally based economy's advancement in coastal areas. At the same time, the digitalized economy has a downward variation behavior in the intensity of carbon emission of each province and has more effects on the intensity of discharge of carbon of eastern and developed regions. Second, the digital economy can remarkably reduce carbon dioxide discharge intensity, and common prosperity is an important mechanism. Third, as it is viewed from the angle of spatial effects, the intensity of carbon dioxide emissions has many effects that are positive in regions, while the digitalized economic impact on the intensity of discharge of carbon dioxide shows a significant negative externality in space. This shows that the intensity of carbon can be effectively moderated by the digitalized economy in the baseline regression test. At the same time, there is still some room for improvement in this paper, mainly due to the limitations of data sources. The evaluation index system of digital economy development level and common wealth is not comprehensive enough, which leads to the final effect measurement results not being comprehensive enough. In future research, we will try to construct a more comprehensive indicator system for the level of digital economy development and common wealth. In addition, exploring the multi-scale regional heterogeneity of the impact of the digital economy on carbon emissions, especially the typical cases, is also a direction worth expanding.

### 7.2. Policy Implications

On the basis of the study conclusions stated above, the paper not only provides strong experiential proof for the impact of the digitally based economic reduction in the discharge of carbon energy but also puts forward the following policy suggestions: First, improve the infrastructure construction related to the digitally based economy and continue to promote the enhancement of the level of digitalized economic advancement. Networked, intelligent, service-oriented, and collaborative infrastructure layers should be continuously promoted in order to fully exploit the benefits of the digital economy in promoting deep decarbonization. Second, introduce differentiated policies for digitally based economic advancement. Intensify efforts to support and develop the digitalization level of underdeveloped regions, promote the widespread and deep penetration of digitalized technology into various fields, stimulate the use and coming up of digitally based technology, its application areas, and models of business in remote and underdeveloped areas, and balance digitally based economic advancement among regions. Third, on the basis of the mechanism of innovation and energy intensity, explore the multidimensional paths through which the digitally based economy promotes carbon emission reduction. The government should create a diversified value transmission and contribution distribution system, systematically guide the development of new employment and entrepreneurship platforms such as diverse social networks, short videos, and knowledge sharing, elevate the innovation level of regions through the digitally based economy, and thereby achieve a reduction in carbon discharge. Additionally, expedite the application of smart energy construction, promote the intelligent upgrading of energy production, transportation, consumption, and other aspects, and drive the low-carbon transformation of the energy industry. Fourth, strengthen

collaborative governance of the discharge of carbon among regions and leverage the spatial "green" radiation impact of the digitally based economy. In addressing the reduction in carbon discharge, it is essential to enhance the exchange and connection of carbon emission policies among regions, establish a unified and joint platform for sharing warnings about regions exceeding emission carbon standards, advance the monitoring of statistics and systems of decision analysis on the basis of recent technologies such as block chain, artificial intelligence, and big data, formulate more flexible and effective policy measures, cultivate a new pattern of digitalized economic governance with diverse governance and coordinated development, and create a low-carbon network of digital economies across multiple regions.

**Author Contributions:** J.G.: Methodology, Validation, Formal analysis, Investigation, Resources. W.Z.: Software, Methodology, Validation, Formal analysis. J.C.: Formal analysis, Writing—reviewing & editing. Z.L.: Resources, Writing—reviewing & editing. All authors have read and agreed to the published version of the manuscript.

**Funding:** This research received no external funding.

**Institutional Review Board Statement:** Not applicable.

**Informed Consent Statement:** Informed consent was obtained from all subjects involved in the study.

**Data Availability Statement:** All data generated or analyzed for this study are included in this published article.

**Conflicts of Interest:** The authors declare that they have no competing interests.

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
