# Peer review of "Digital Economy Development, Common Prosperity, and Carbon Emissions: An Empirical Study in China"

_economies, doi:10.3390/economies12050120_

Round 1
Reviewer 1 Report
Comments and Suggestions for Authors
In the study, the authors addressed an important and current issue of the relationship or interdependence of the development of the digital economy and the scale of carbon dioxide emissions. They chose China as the research location. They provided an interesting justification for taking up the topic and then presented a review of the literature available in this area. They put forward three research hypotheses regarding the relationship between the development of the digital economy and energy consumption, the scale of emissions or reduction of greenhouse gas emissions, general prosperity, and spatial aspects of carbon dioxide emissions. The hypotheses are formulated correctly, and the adopted research method is also correct. Based on relevant data from 2010 - 2020, they built an econometric model for 30 provinces of China. The model is methodologically correct.
As a result of the research, the authors found that the digitalization of China's economy was relatively fast, and the greatest benefits were achieved in the most developed coastal areas. The digital economy resulted in a decrease in behavioral diversity in terms of carbon intensity in the provinces studied, including affecting emission intensity in eastern and developed regions. Spatially, the effects vary, but are generally positive. What is somewhat new here is the statement about a significant negative external effect in space. From a practical perspective, it is important to recognize that carbon intensity can be regulated by the digital economy.
In the final part, the summary, the authors point to the need to build infrastructure necessary for the use of digital technology and promote digital economic development. They emphasize the need to pursue a diversified policy for economic development, i.e. supporting the digitalization of less developed regions, applying digital technology in various fields, stimulating the use and creation of digital technologies and balancing the dissemination of digital technology between regions. Overall, the study is interesting and expands knowledge on the impact of the digitization of the economy on its development and the level of carbon dioxide emissions.
Author Response
Dear editor:
I am very glad to receive your reply. Thank you very much for your support of this article.
I also continued to make some modifications to the paper, making the abstract of the paper more concise and focused; The hypothesis part of the paper is further enriched; Enriched the references and other minor modifications.
Wish you a happy life.
Best Regards!Sincerely yours, the Author

Reviewer 2 Report
Comments and Suggestions for Authors
-This submission has not sufficiently clarified the novelty and meaning of the proposed analysis. The analytical discussions section should be improved.
-The originality of the work should be given as a bullet point. For example; The main contributions of the current study to the literature are listed below
-For emphasizing the originality of the study, a new table of related studies will be added in the related section. A comparison table can be added according to the features. For example; Comparison of works regarding systems.
-It is not clear how the indirect effect (hypotheses 2) part happens.
-There was a problem with the iThenticate report. Table-2 should be rewritten with all its subtexts. Reference should also be given.
Author Response
Dear editor:
I am very glad to receive your reply. Thank you very much for your support of this article.
The following changes were made to the article:
- The paper further adds the significance of studying the impact of digital economy development on carbon emission reduction in the introduction section.
- The measurement indicators of digital economy development are highlighted in the contribution of the paper.
- Further clarified the formulation of hypothesis 2.
- The source of the indicator system according to Table 2 is also provided for reference.
Wish you a happy life.
Best Regards!
Sincerely yours, the Author

Reviewer 3 Report
Comments and Suggestions for Authors
Thank you for the opportunity to review this manuscript. The study examines the impacts of the digitally based economy on energy consumption and carbon dioxide emissions.
Some suggestions to consider when reviewing this paper:
- The abstract succinctly presents the main findings, initially highlighting the direct correlation between the digital economy and energy-related carbon dioxide emissions. It proposes an econometric model based on panel data from 2010 to 2020 from 30 provinces in China. Additionally, common prosperity plays a mediating role in mediating the impact of the digital economy on reducing energy-related carbon dioxide emissions. I recommend removing the expressions "primarily" and "secondly" from the abstract and incorporating details about the methodology used (cross-sectional study, regression analysis, tests, software used, etc.) while emphasizing the study's originality.
- In the "Introduction" section, the negative effect of CO2 emissions on human life, economic growth, and the globe is presented. Therefore, Chinese public institutions must carefully consider the effects of digital economic progress on reducing CO2 emissions in their future development strategies. I recommend defining the decision problem: what questions does your study aim to answer? Then, before presenting the paper's contributions, the purpose and secondary objectives of your study should be clearly and concisely presented.
- In the "Research on carbon emissions" section, the analysis provides a comprehensive overview of existing research related to the determinants and contributors to carbon emissions growth, the potential of new digital technologies to reduce carbon emissions, identifying effects of the digital economy on carbon emissions in neighbouring areas, the role and importance of common prosperity in mediating the relationship between digital economic impact and increased energy-related carbon discharge, as well as the potential effects of digital economic progress on carbon emissions. However, I emphasize that there is a lack of or few studies in the "Research on carbon emissions" section that present the main issues/gaps/limitations related to the analyzed problem. Don't forget, the study objectives should be in line with the research hypotheses.
- The “4. Model and data" section presents the methodology for selecting explained, explanatory, intermediary, and control variables. The study employs Baseline Regression Test and Mediation Mechanism Test for testing and validating the statistical hypotheses. Data sources could be enhanced by highlighting other studies that use similar models and data sources and by highlighting the advantages offered by this type of data collection and selection.
- The "5. Empirical results and analyses" and “6. Discussion” sections are well-grounded, presenting and confirming the obtained results as well as those from existing studies.
- In the "Conclusions and policy implications" section, it is essential to add the research limitations, scientific implications (for researchers, students, etc.), and future research directions.
Author Response
Dear editor:
I am very glad to receive your reply. Thank you for your valuable input!
The following changes were made to the article:
- Simplified and focused summary section.
- The introductory section is also focused and designed to respond to what is wrong.
- in the conclusion section of the article also added the limitations of the study.
- data sources have been added in the fourth section.
Wish you a happy life.
Best Regards!
Sincerely yours, the Author

Reviewer 4 Report
Comments and Suggestions for Authors
ID: economies-2978161
Title: Digital economy development, common prosperity and carbon emissions: An empirical study in China
Using the passive voice instead of -we our makes your study more academic.
-Abstract should be shorter and clearer.
-The introduction section should include studies discussing China's environmental problems: Determinants of the load capacity factor in China: a novel dynamic ARDL approach for ecological footprint accounting. Resources Policy; Is reducing fossil fuel intensity important for environmental management and ensuring ecological efficiency in China?. Journal of Environmental Management
-Research gap should be stated in detail at the end of the literature.
-The econometric method used in the study should be presented in more detail.
-It is essential to enrich your work by taking advantage of current studies on digitalization and the environment: Environmental sustainability in the OECD: The power of digitalization, green innovation, renewable energy and financial development. Telecommunications Policy; Digital economy and carbon dioxide emissions: examining the role of threshold variables. Geoscience Frontiers; he role of ICT, R&D spending and renewable energy consumption on environmental quality: Testing the LCC hypothesis for G7 countries. Journal of Cleaner Production
-A separate section should be opened as the Future research section and limitations and suggestions should be discussed there.
-How is your work different from previous studies? Many studies have been conducted on China's digitalization and CO2 emissions. In the Discussion section, emphasize the difference between your work and the previous ones.
Author Response
Dear editor:
I am very glad to receive your reply. Thank you for your valuable input!
The following changes were made to the article:
- Simplified and focused summary section.
- Appropriate references have also been added in the introduction section.
- In the conclusion section of the article also added the limitations of the study.
- Further enriched the references of the article.
5.A description of the innovations has also been added in the contribution of the article.
Wish you a happy life.
Best Regards!
Sincerely yours, the Author

Round 2
Reviewer 2 Report
Comments and Suggestions for Authors
The table regarding the 3rd comment has not been added. Other comments have been completed. My opinion is minor revision.
-For emphasizing the originality of the study, a new table of related studies will be added in the related section. A comparison table can be added according to the features. In order to emphasize the originality of the study.
Author Response
Dear editor:
It is a pleasure to receive your reply. Thank you for your valuable comments! I have made changes according to your comments. Five articles on digital economy indicator systems are listed in the literature review section as evidence of the originality of the article's research.
Wish you a happy life.
Best Regards!
Sincerely yours, the Author

Reviewer 4 Report
Comments and Suggestions for Authors
The authors made the all suggestions properly.
The article can be published with this revised version.
Author Response
Dear editor:
I am very glad to receive your reply. Thank you for your valuable input! In the next step of my academic path, I will be more deeply involved in this field of study.
Wish you a happy life.
Best Regards!
Sincerely yours, Jingke Gao